

# Effects of mental rotation on map representation in orienteers—behavioral and fNIRS evidence

Mingsheng Zhao[1,*], Jingru Liu[2,*], Yang Liu[1] and Pengyang Kang[1]

[1] School of Physical Education, Shaanxi Normal University, Xi'an, Shaanxi, China
[2] Physical Education Department, Xi'an University of Posts and Telecommunications, Xi'an, Shaanxi, China
* These authors contributed equally to this work.

## ABSTRACT

**Objective:** Taking orienteering as an example, this study aimed to reveal the effects of mental rotation on orienteers' map representation and their brain processing characteristics.

**Methods:** Functional near-infrared spectroscopic imaging (fNIRS) was used to explore the behavioral performance and cortical oxyhemoglobin concentration changes of map-represented cognitive processing in orienteering athletes under two task conditions: normal and rotational orientation.

**Results:** Compared to that in the normal orientation, athletes' task performance in the rotated orientation condition was significantly decreased, as evidenced by a decrease in correct rate and an increase in reaction time; in the normal orientation condition, blood oxygen activation in the dorsolateral prefrontal lobe was significantly greater than that in the ventral prefrontal lobe, which was significantly correlated with the correct rate. With rotating orientation, the brain oxygen average of each region of interest was enhanced, and the brain region specifically processed was the ventral prefrontal lobe, specifically correlating with the correct rate.

**Conclusions:** Mental rotation constrains the map representation ability of athletes, and map representation in rotational orientation requires more functional brain activity for information processing. Ventral lateral prefrontal lobe activation plays an important role in the map representation task in rotational orientation.

## INTRODUCTION

The ability of exercise and cognitive training to improve cognitive performance has been demonstrated in many studies (*Waddington Emma & Heisz Jennifer, 2023*), but the preintervention of orienteering may be the most effective since it resembles the hunter-gatherer activities that humans have engaged in over the course of their evolution while combining high-intensity intermittent exercise with navigation (*Kolb, Sobotka & Werner, 1987*). Studies have demonstrated that professional orienteers exhibit better spatial navigation and memory abilities and that the navigation features specific to

Corresponding authors
Yang Liu, liuyang0330@snnu.edu.cn
Pengyang Kang,
bobby197896@163.com

orienteering may delay age-related dementia. These findings suggest that combining running with spatial navigation may be the best way to exercise the brain and effectively protect against cognitive decline (*Waddington Emma & Heisz Jennifer, 2023*). Interventions through orienteering have been demonstrated effective in arresting cognitive decline.

Map representation is important for orienteering athletes to complete the competition (*Kolb, Sobotka & Werner, 1987*). The athletes' ability to quickly recognize the information of map symbols and match it with the real information of the real-world environment has been examined (*Mottet & Saury, 2013*; *Newton & Holmes, 2016*). The map information must be analyzed considering existing knowledge and experience. This specific process involves extracting features of map symbols, integrating and processing them, and synchronizing and matching them with information about the external environment to calibrate the map and find the target. The whole process involves multiple spatial cognitive processing processes, such as attention, memory, and mental rotation (*Liu & He, 2017*; *Weronika et al., 2019*). During the map representation process, the orientations of the map and the field typically differ, and due to the constant change in rotational orientation, people must continually adjust the positional relationship between the map and the reference object (*Fang et al., 2019*) and match the information in the real world. The brain needs to represent the map to form a mental image map, which requires good visual search (*Liu, 2019b*) and mental rotation (*Uttal et al., 2013*) abilities to assist athletes in spatial localization to accurately and quickly match the real scene. *Eccles, Walsh & Ingledew (2002)* proposed using the rooted theory approach that orienting maps, real scene information and visual attention during travel are core factors in the program. The efficiency of scene recognition is affected by rotational orientation (*Liu & He, 2016*), which in turn affects the efficiency of route decision-making, and the ability to mentally rotate is strongly correlated with task performance (*Song, Tang & Xian, 2021*; *Yi et al., 2022*), which constrains map literacy and spatial localization. Therefore, exploring the effects of mental rotation on the cognitive processing of map representation in orienteering athletes provides theoretical support for deliberate training to improve athletes' map representation ability in this program.

Functional near-infrared spectroscopic imaging (fNIRS) is an emerging technology for functional brain imaging that utilizes a near-infrared light source capable of penetrating human tissues to detect changes in the concentration of $HbO_2$ and HbR, the major absorbers of near-infrared light (*Ferrari & Quaresima, 2012*; *Kopton & Kenning, 2014*). fNIRS can indirectly quantify neural activity, providing technical support for the present study. Brain imaging technology has unique advantages, such as high safety, portability, high spatial and temporal resolution, and less influence by head movements (*Yücel et al., 2017*; *Pinti et al., 2020*) and has been widely applied in research into sports, including soccer (*Liu, 2019a*), badminton (*Chen, 2017*) and tai chi (*Han, 2019*). The prefrontal cortex area (PFC) is the most anterior region of the frontal cortex, accounting for approximately half of the frontal lobe, and is located in the area before the central sulcus and above the lateral sulcus. The PFC is closely connected with the parietal lobe, occipital lobe, temporal lobe and other brain regions. In the information-processing activity of the brain involving

multiple pieces of information from various brain regions is summarized in the PFC, which performs the final processing and integration of processing (*Zhou, 2008*). This is the area responsible for higher cognitive activities, such as motivation, problem solving, thinking and judging, and making plans (*Koechlin et al., 1999*; *Mandrick et al., 2013*; *Miller & Cohen, 2001*; *Robert & Tirin, 2011*). Therefore, PFC was selected as the main functional area in this study. Based on the 3D localization results in existing studies, the following four areas of interest were delineated: dorsolateral prefrontal (DLPFC), frontal pole area (FOA), ventral lateral prefrontal (VLPFC), and box frontal area (OFA) (*Ferrari & Quaresima, 2012*). To date, orienteering cognitive studies have focused on behavioral testing of cognitive indicators such as visual attention (*Liu & He, 2017*), working memory (*Liu, 2021*), and mental rotation (*Song, Tang & Xian, 2021*). Practice interventions through orienteering have all been found to have some intervention benefits on the cognitive abilities of primary and secondary school students, children with ADHD, people with intellectual disabilities, and elderly individuals (*Song et al., 2020*; *Yang & Yang, 2018*; *Tang & Liu, 2021*; *Bao, Wei & Liu, 2021*). However, previous studies lacked the effect of mental rotation on the effects and neural mechanisms explored on map representation in orienteering athletes.

To address the above-listed issues, the present study further explored the brain processing mechanisms in the map representation task of orienteering athletes with the help of fNIRS technology. The following hypotheses were proposed: (1) the behavioral performance of map representation differed significantly under different mental rotation conditions; (2) the blood oxygen response pattern of each region of interest was affected by mental rotation factors during map representation; and (3) the response pattern of the prefrontal lobe of the brain differed with the rotation of the map direction during orienteering sport-specific practice. The function of prefrontal regions can undergo specific changes after prolonged practice and improve the cognitive ability of the group.

## MATERIALS AND METHODS

### Experimental subjects

Thirty-three (15 male, age 20.16 ± 1.13, 18 female, age 19.83 ± 0.57) varsity orienteering players from a university with 3.16 ± 0.56 years of training were selected for the experiment. The following inclusion criteria were used for all participants: (1) consistent education; (2) right-handed; (3) normal vision; (4) no history of any neurological disease; (5) able to master the specific skills of orienteering relatively proficiently; and (6) not having participated in similar experiments before. Remuneration was given upon completion of the experiment. The informed consent of all participants was obtained and documented (see Supplemental Materials). The study was approved by the ethics and morality committee of Shaanxi Normal University (Approval number: SNNU2023301).

### Experimental design and materials

A 2 × 4 two-factor mixed design was used. Factor 1 was the task condition (normal and rotational orientation); Factor 2 was the brain regions of the prefrontal lobe: dorsolateral prefrontal lobe (DLPFC), frontal polar region (FOA), ventral lateral prefrontal lobe
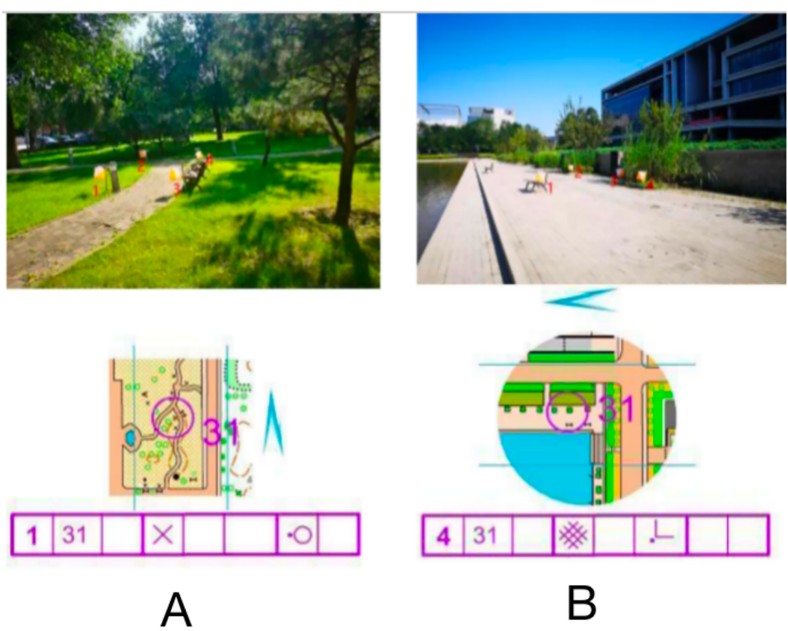

**Figure 1 Examples of maps with different orientations of stimulus materials.** Normal orientation (A); rotation orientation (B).                                   

(VLPFC), and orbitofrontal region (OFA). The dependent variables were as follows: behavioral indicators (correctness, reaction time) characterized by different rotational position maps of the subjects and the concentration of oxyhemoglobin ($HbO_2$) in each brain region of the prefrontal lobes.

The stimulus material consisted of an orienteering map and a corresponding real-world photograph, including map information, a checkpoint description sheet (detailed information describing the location of points in the map), a pointing sign, and a real-world photograph. The map was divided into normal orientation (consistent orientation) and rotated orientation (inconsistent orientation) according to the degree of matching with the orientation of the real-world photograph, and there were four choice locations in the real-world photograph (indicated by white and orange dot marker flags), one of which was the correct option consistent with the map dot location. As shown in Fig. 1A, the map point number is 31, the checkpoint description table indicates on the left side of the special feature, the north pointing marker shows the map orientation consistency, and the correct option in the live photograph should be point 1. All stimulus materials of the experiment were produced, screened and proofread by three orienteering specialists.

## Experimental equipment

A portable functional near-infrared spectroscopic imager (LIGHTNIRS system) manufactured by Shimadzu Corporation, Japan, was used to monitor the oxyhemoglobin ($HbO_2$), deoxyhemoglobin (HbR), and total blood oxygen concentrations. $HbO_2$ is more sensitive to local blood flow changes in the brain than HbR; therefore, $HbO_2$ was selected to reflect the level of neural activation in the brain in this study (*Wen & Wang, 2016*; *Skau et al., 2022*; *Fan et al., 2021*).

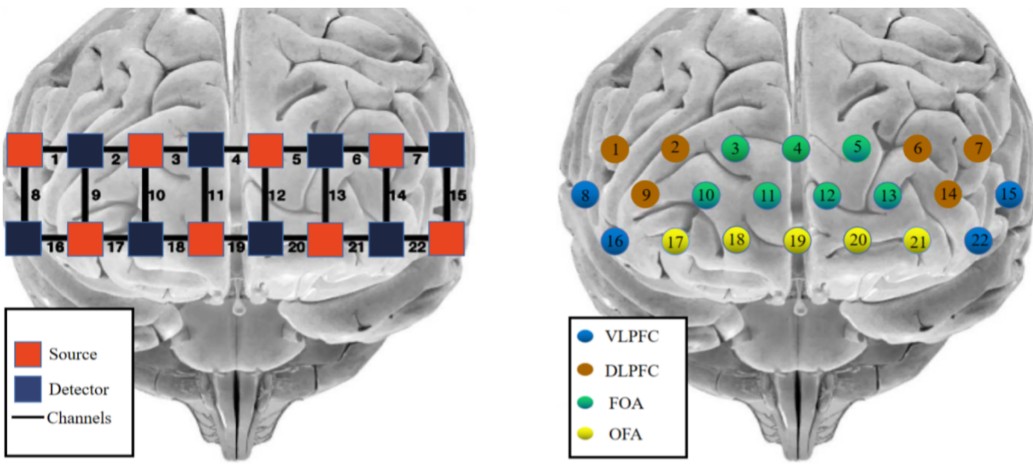

**Figure 2 fNIRS channel layout and calibration brain region information.** Red indicates emitter-signal source, blue indicates detector-detector, and numbers indicate-channel.

The photopolar probe was located in the prefrontal lobe (PFC), and the lowest probe was placed along the Fp1–Fp2 line using the international 10–20 localization system as a reference, using the system's own PFC template, with a multichannel connected layout of 2 × 8 photopolar probes (eight transmitting and eight receiving photopoles), forming a total of 22 channels, with two similar photopoles lined up at intervals and a distance of 3 cm between adjacent photopoles. The MNI coordinates of each channel position were determined by the NIRS_SPM software spatial probability alignment method, and the corresponding brain areas were found in the adult Brodmann area (Brodmann) atlas. The calibration information is shown in Fig. 2.

## Experimental procedure

Before the experiment, the participants were allowed to familiarize themselves with the experimental environment, and relevant information such as gender, age, training duration, and exercise level was recorded. Participants were informed of the experimental precautions during the experiment and instructed by the experimental staff to wear fNIRS optical polar caps adjusted to the appropriate looseness, fixed with nylon buckles and positioned for calibration. Then, the optical fiber was inserted into each probe to check whether the light emission and reception of each channel were normal, and after the signal of each channel was stabilized, a zero reset was performed and the measurement was initiated. Two tasks were included for testing (normal orientation and rotational orientation) with the same operational procedure.

Experimental stimuli were prepared using the neuropsychological programming software "E-prime 3.0" and consisted of a practice phase and a formal experiment. First, the instructions were presented on the entire computer screen, and the participants were asked to familiarize themselves with the task and the procedure and press the space bar to practice. Then, a gaze point was presented for 1 s, followed by the stimulus picture, and the participants had to carefully observe the picture information, select the option with the

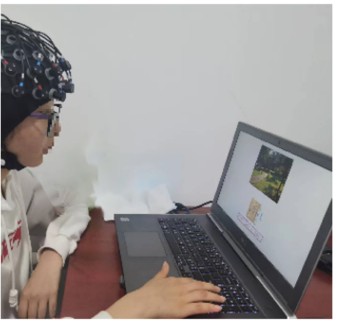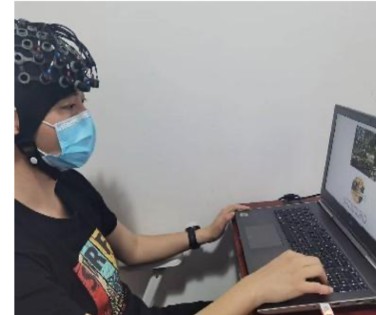

**Figure 3** **fNIRS testing process.**

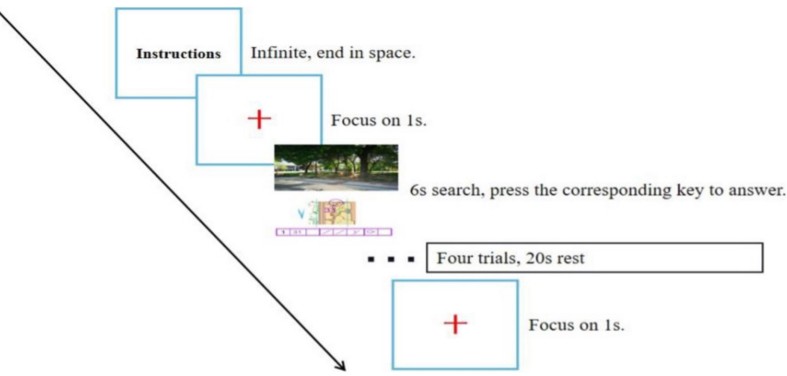

**Figure 4** **Flow chart of the experiment.**

same position in the real photograph according to the map orientation and point location, and press the corresponding numeric key according to the checkpoint description sheet. Participants were given 6 s to make a judgment, and feedback on the results was given after the judgment was completed. The test process is shown in Fig. 3.

In the formal test phase, the resting-state electrophysiological signal was first collected for 30 s with the participant in a relaxed state. Then, the test task started using the same process as in the practice phase, but the participant did not receive feedback after making a judgment, and the stimulus was presented cyclically until the end of the test. Every four trials constituted a block, six blocks in total, with a rest of 20 s between each block (to ensure that the relative concentration of cerebral blood oxygen in the PFC returned to the baseline value), for a total of 24 trials (as shown in Fig. 4). The program automatically recorded the reaction time and correct rate of the participant performing the task, and the NIR device collected cerebral blood oxygen data.

## Data collection and analysis
### Behavioral data
SPSS 26.0 (SPSS Inc., Chicago, IL, USA) was used to test the normal distribution of the measurement data, which was greater than the 0.05 threshold, indicating that it was normally distributed; the correct rate and response time of the subjects' map
representations under two different rotational orientations were analyzed by ANOVA, and the differences in behavioral indices under different rotational orientations were observed.

### fNIRS data

The raw data collected were analyzed using fNIRS device software. Based on the NIRS_SPM software on the MATLAB (R2013b) platform (MathWorks, Natick, MA, USA), the light intensity data were converted to blood oxygen data using the modified Beer–Lambert law and preprocessed to eliminate outliers and improve the signal-to-noise ratio; thus, the overall filtered signal was easy to analyze for the subsequent calculations. These calculations included the alignment of MNI coordinates, construction of a design matrix based on the general linear model (GLM), a low-pass filter based on the hemodynamic response function (HRF) with time derivatives, and a high-pass filter based on the discrete cosine transform (DCT) detrending algorithm; then, the beta values under the task conditions were evaluated as an indicator of activation of the corresponding channel. Finally, the beta of each channel contained in the region of interest (ROI) was averaged, and this mean value represented the activation strength of that region of interest (ROI) (*Huang et al., 2020*; *Lei et al., 2021*). The cerebral oximetry data were tested for normal distribution using SPSS 26.0, and the Shapiro–Wilk test (S–W) showed that the values exceeded the 0.05 threshold, indicating normal distribution; a two-task condition (normal orientation, rotated orientation) × 4 (VLPFC, DLPFC, FOA, OFA) two-factor repeated-measures analysis of variance (ANOVA) was performed, and statistics that did not satisfy the assumption of sphericity were calibrated using Greenhouse's method. Thus, the quantities were corrected by Greenhouse's method, and further tests were corrected for multiple corrections using Bonferroni's method, with $P < 0.05$ considered to indicate significance. Bivariate correlations between behavioral and cerebral blood oxygenation data were analyzed by GraphPad Prism 8.0 (GraphPad, San Diego, CA, USA) for (1) correct rate of normal orientation map representation with cerebral blood oxygenation (four brain regions of interest) and (2) correct rate of rotational orientation map representation with cerebral blood oxygenation (four brain regions of interest).

## RESULTS

### Behavioral data results

The descriptive statistics of correctness and response time for map representations with different rotational orientations are shown in Table 1.

The Shapiro–Wilk results showed that correctness ($P = 0.210 > 0.05$) and response time ($P = 0.395 > 0.05$) were normally distributed. The analysis of variance (ANOVA) revealed a significant difference in correctness for the map representations, with the value for normal orientation significantly higher than that for rotational orientation $F(1, 62) = 4.223$, $P = 0.044$, $\eta^2 = 0.064$. Response times for map representations varied significantly, with those in the normal orientation significantly shorter than those in the rotational orientation $F(1, 62) = 4.620$, $P = 0.036$, $\eta^2 = 0.069$.

**Table 1 Results of behavioral indicators of map representation.**

|  | Correct rate | Reaction time (ms) |
|---|---|---|
| Normal orientation | $0.67 \pm 0.053^{*}$ | $3{,}718.285 \pm 346.790^{**}$ |
| Rotational orientation | $0.54 \pm 0.065^{*}$ | $3{,}910.818 \pm 369.414^{**}$ |

Notes:
$^{*}$ Represents $0.001 < P < 0.05$.
$^{**}$ Represents $P \leq 0.01$.

**Table 2 HStatistical results of $bO_2$ descriptions for map-represented regions of interest in the brain.**

|  | DLPFC | FOA | VLPFC | OFA |
|---|---|---|---|---|
| Normal orientation | $-0.65 \pm 2.61^{**}$ | $-1.19 \pm 2.80$ | $-2.22 \pm 4.71^{**}$ | $-1.83 \pm 6.10$ |
| Rotational orientation | $1.26 \pm 2.76^{**}$ | $-0.46 \pm 2.52$ | $1.42 \pm 3.67^{**}$ | $-0.97 \pm 3.46$ |

Notes:
$^{*}$ Represents $0.001 < P < 0.05$.
$^{**}$ Represents $P \leq 0.01$.

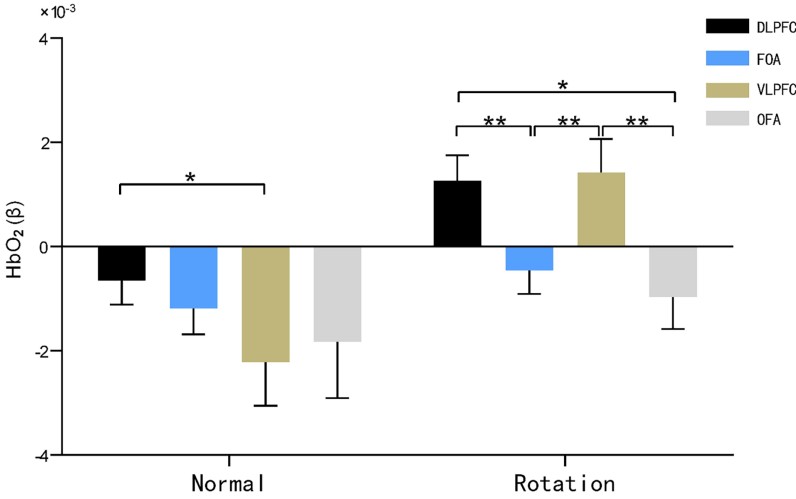

**Figure 5 HbO$_2$ results of brain regions of interest characterized by maps.** Note: $^{*}$ $0.001 < P < 0.05$; $^{**}$ $P \leq 0.01$.

## fNIRS data results

A two-factor repeated-measures ANOVA was performed on the beta values of cerebral blood oxygen (HbO$_2$) of the subjects in the two-task condition (normal orientation, rotational orientation) × 4 brain regions (DLPFC, FOA, VLPFC, OFA) using rotational orientation and brain region of interest as independent variables, and the results are shown in Table 2 and Figs. 5 and 6 below.

The main effect of brain area was not significant (F(3, 29) = 2.027, $P = 0.126$, $\eta^2 = 0.177$), and the main effect of rotational orientation was significant F(1, 31) = 13.367, $P = 0.001 < 0.05$, $\eta^2 = 0.177$. The simple-effects analysis indicated that the interaction between brain area and rotational orientation was significant F(3, 29) = 3.333, $P = 0.033$, $\eta^2 = 0.256$.

**Figure 6 Activation map of HbO$_2$ values of the map-characterized brain.** Normal orientation (A); rotational orientation (B).                     

In the dorsolateral prefrontal lobe (DLPFC) and ventral lateral prefrontal lobe (VLPFC), significant differences in oxygen activation emerged between the two rotational orientations, as demonstrated by significantly increased oxygen activation in the rotational orientation than in the normal orientation, with F values of 11.330 ($P = 0.002$, $\eta^2 = 0.268$) and 13.487 ($P = 0.001$, $\eta^2 = 0.303$), respectively.

In the normal orientation condition, blood oxygen activation was significantly greater in the dorsolateral prefrontal lobe (DLPFC) than in the ventral lateral prefrontal lobe (VLPFC) ($P = 0.044$, $\eta^2 = 0.139$); in the rotational orientation condition, there was no significant difference in blood oxygen activation in the dorsolateral prefrontal lobe (DLPFC) and ventral lateral prefrontal lobe (VLPFC) ($P = 0.851$, $\eta^2 = 0.324$), and both brain regions showed significantly increased activation in comparison with the frontopolar area (FOA) ($P = 0.004$, $P = 0.010$) and orbitofrontal area (OFA) ($P = 0.014$, $P = 0.004$). This result suggests that the ventral lateral prefrontal lobe (VLPFC) becomes more activated by blood oxygen in the map representation task in rotational orientation.

## Correlation analysis of correctness and activation intensity of brain interest areas

Pearson's correlation analysis of behavioral performance (correct rate) of two different orientation map representations with cerebral blood oxygen (HbO$_2$) β-values of different brain regions revealed that the correct rate of normal orientation significantly positively correlated with the dorsolateral prefrontal lobe (DLPFC) ($P = 0.04 < 0.05$, r = 0.35), and the correct rate of rotational orientation had a significantly positively correlated with the dorsolateral prefrontal lobe (DLPFC) ($P = 0.04 < 0.05$, r = 0.36) and ventral lateral prefrontal lobe (VLPFC) ($P = 0.03 < 0.05$, r = 0.38). The experiments illustrate that as the orientation is rotated, the activation level of the VLPFC rises, the correctness of the map representation increases, and the enhanced activation of the VLPFC correlates with the ability to mentally rotate, consistent with the results of previous studies (Table 3 and Fig. 7).

**Table 3 Correlation results between fNIRS and behavior (Pearson correlation coefficient r).**

| Indicators | Correlation | DLPFC | FOA | VLPFC | OFA |
|---|---|---|---|---|---|
| Normal orientation correct rat | Pearson correlation | 0.404* | 0.106 | 0.295 | 0.280 |
| | Significance (two-tailed) | 0.022 | 0.562 | 0.105 | 0.121 |
| Rotation orientation correct rate | Pearson correlation | 0.356* | 0.262 | 0.494** | 0.050 |
| | Significance (two-tailed) | 0.045 | 0.148 | 0.004 | 0.784 |

**Notes:**
  * Significant correlation at the 0.05 level.
  ** Significant correlation at the 0.01 level.

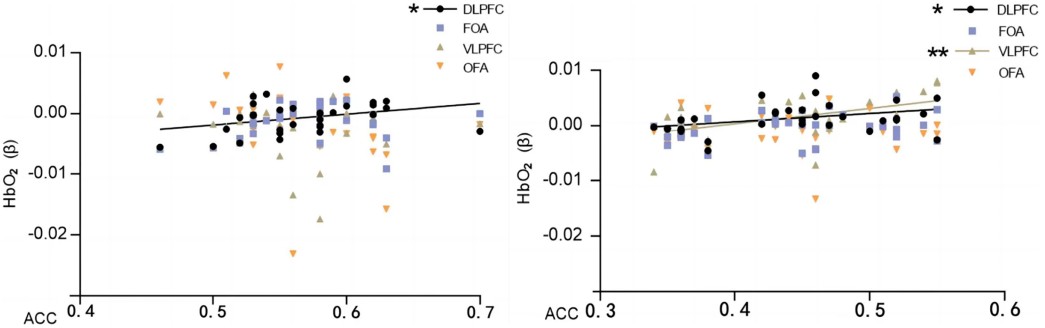

**Figure 7 Map characterizing the correlation between $HbO_2$ values and correctness in brain regions of interest.** Normal direction on the left; rotational direction on the right. Note: * $0.001 < P < 0.05$; ** $P \leq 0.01$.

The results showed that the correct rate of normal orientation was significantly and positively correlated with the dorsolateral prefrontal lobe (DLPFC) ($P = 0.022 < 0.05$, $r = 0.404$) and not with any other brain regions; the correct rate of rotational orientation was significantly and positively correlated with the dorsolateral prefrontal lobe (DLPFC) ($P = 0.045 < 0.05$, $r = 0.356$) and ventral lateral prefrontal lobe (VLPFC) ($P = 0.004 < 0.05$, $r = 0.494$) and was significantly and positively correlated with the frontopolar area (FOA) and orbitofrontal area (OFA) but not with the orbitofrontal area (OFA).

# DISCUSSION

## Behavioral characteristics of map representations

TA characterization of orienteering athletes' mental representations of maps in different orientations revealed significant differences in behavioral performance, as evidenced by significantly lower rates of correctness and significantly higher reaction times for rotationally oriented maps compared to normal orientations. This result is consistent with previous findings (*Bethell-Fox & Shepard, 1988*). The general process of map representation is as follows: inputting map information–processing and analyzing map information (perception, attention, memory)–matching map to match with the real scene–making a judgment. In the normal orientation map condition, due to the consistent orientation of the map, athletes only need to input information and process information to make a judgment. In contrast, in the rotated orientation map condition, participants needed to recharacterize the map to match real-scene features after analyzing and

processing the map with mental rotation, and the brain required more time to consider the correct reference point information, making the recognition processing consume more cognitive resources (*Pan, Jiao & Jiang, 2014*). Thus, the reaction time was increased, and the correctness was lowered.

The process of recognizing the direction and position of an object during orientation determination by athletes requires cognitive processing of the object. In normal orientation, the task is simpler, and the allocation of resources in the relevant brain regions for spatial cognitive processing is easier for the individual, with less difficulty in mobilizing attention resources (*Wu et al., 2013*). In contrast, rotational orientation requires more cognitive effort to complete the task. According to cognitive load theory, an individual's cognitive resources are limited, and when the difficulty or number of tasks increases, the individual's cognitive load increases, and the relevant cognitive resources that must be extracted by the individual increase significantly (*Sweller, 1988*). According to the findings of this study, the correctness rate of the map of the rotational orientation was only $0.54 \pm 0.065\%$, while the response rate was only $0.54 \pm 0.065\%$, while the response time was $3,910.818 \pm 369.414$, consistent with previous findings indicating that the complexity of the information processing process consumes excessive attention resources and reduces the correct rate (*Zhang, 2020*; *Hou et al., 2018*). Therefore, the rotational direction map characterization findings suggest that the increase in mental rotation can increase the orienteering athletes' cognitive load and mobilize more attention resources to complete spatial cognitive tasks.

## Brain activation mechanisms of map representations

The fNIRS results showed that the activation intensity of all regions of interest in the prefrontal lobe (PFC) was enhanced in the rotational orientation condition, which was attributed to the fact that the change in rotational orientation made the task more difficult and required participants to pay more attention, which supplied the prefrontal lobes of the brain with more energy substances to make the cerebral load larger; therefore, the activation of cerebral blood oxygenation increased (*Zhao et al., 2022*; *Zeki et al., 1991*), consistent with previous fNIRS studies (*Wang et al., 2021*). The cognitive processing pathway hypothesis states that cognitive processing mainly consists of two different cortical processing pathways (*Zeki et al., 1991*): the ventral pathway (what pathway) and the dorsal pathway (where pathway). The dorsolateral prefrontal cortex (DLPFC) and the ventral lateral prefrontal cortex (VLPEC), as the main functional areas of the prefrontal cortex, are closely related to brain functions related to motor cognition. The present study shows that there are differences between different brain regions under equal conditions. Under both normal and rotational orientation conditions, the differences in blood oxygen activation between DLPFC and VLPFC brain regions were significant, whereas no significant differences were found between FOA and OFA brain regions, potentially due to the differences in prefrontal brain regions with different functions. Moreover, the negative activation in the FOA and OFA brain regions may result from anticorrelation between the activity of the default network and the activity of the attention network (*Wang et al., 2021*).

*Kelly & Garavan (2005)* suggested that the degree of involvement of the DLPFC depends on the cognitive characteristics of the task performed, which is the main functional area for spatial information retention, monitoring, and cognitive decision-making (*Chen, 2015*; *He, 2017*). The map representation process involves recognizing, coding, and representing map information as real-world information and is highly demanding of athletes' spatial cognitive abilities. Therefore, under normal orientation conditions, activation in the DLPFC brain regions was greater than that in other brain regions. *Hikosaka et al. (2002)* noted that the DLPFC involves spatial location acquisition, processing of initial perceptual inputs and spatial sequence depictions and that spatial information in rotational orientation requires more spatial sequence depictions and more cognitive effort to be invested; thus, the concentration of oxyhaemoglobin in the DLPFC brain regions is increased, *i.e.*, the cerebral blood oxygen activation intensity increases. Another study demonstrated that when disrupting the excitation of the DLPFC brain region, participants showed increased response times in spatial information processing tasks, and the DLPFC was also significantly activated when the complexity of the task process and the need for integration became greater (*Hoshi, 2006*). The results of the present study also show that the activation of the DLPFC brain region is significantly greater in the rotational than in the normal orientation, further validating the spatial cognitive function of this brain region in orienteering-specific tasks.

Activation of VLPFC brain regions was significantly greater in the rotated orientation condition than in the normal orientation condition. This may be because rotational orientation requires subjects to make mental rotations to represent and manipulate the spatial relationships of objects. In other words, a rotated visual representation is constructed in the brain to match the "north" of the map with the "north" of the actual scene, increasing the cognitive processing required for the task. Previous studies have also demonstrated that the VLPFC brain region, together with other related regions (*i.e.*, angular and cingulate gyrus, posterior pleural cortex), constitutes the so-called "default mode network" and suggests visions of the future and memory retrieval, which are closely related to human navigational processes, are involved in controlling functions such as orientation cognition and map representation, and are key for spatial attention and scene memory (*Buckner, Andrews-Hanna & Schacter, 2008*; *Boccia et al., 2017*; *Wilson Charles et al., 2007*). In the rotational orientation condition, orienting athletes must rotate themselves in the environment or consider the environment as an overall rotating external entity, requiring them to reorient themselves spatially, increasing task difficulty. Thus, the VLPFC brain region is significantly activated as a major functional area with some advantageous processing.

In summary, the map characterization process includes inputting, encoding, storing, extracting and matching the map information to the real world. The DLPFC was significantly activated regardless whether the task was simple or complex or if integration demands were increased. In the rotational condition, the change in map orientation may enhance the network connection between brain regions, requiring more brain regions to cooperate and complete the map representation task. The VLPFC is a key region for spatial

attention, scene memory, and spatial localization; thus, it was only significantly activated in the rotational (complex) condition.

The predominance of activation in FOA and OFA brain regions remains unclear. Various hypotheses exist. It has been suggested that with the onset of aging, there is a gradual decline in cognitive ability and a significant slowing of cognitive processing speed, that older adults face higher complexity than younger adults when performing the same tasks, that the activation of the FOA brain region is significantly enhanced, and that older adults presumably use compensatory strategies to accomplish more complex information processing, demonstrating a compensatory mechanism for the brain's processing processes (*Rypma, Eldreth & Rebbechi, 2007*). The theory of prefrontal function also suggests that the information processing process goes from backward to forward in this region, while the frontal pole region is at the top of this hierarchy, and the frontal pole region is recruited to play a functional role only when the low-functioning system fails (*Shi, Wang & Gu, 2014*). The OFA brain region mainly modulates some high-level cognitive functions, such as neural activities of reward judgment, working memory, risk assessment, and emotional adjustment, which individuals rely on to avoid risks (*Rudebeck & Rich, 2018*; *Stalnaker, Cooch & Schoenbaum, 2015*). In this study, there were no significant differences in the activation of the FOA and OFA brain regions under both orientation tasks.

In this study, the difference between the dorsal and ventral PFC was also confirmed by map representation in orienteering athletes with different rotational orientations, supporting previous results. This study identified the important roles of DLPFC brain regions and VLPFC brain regions in maps, providing an objective basis for us to improve athletes' map representation ability through orienteering training in the future. In the future, specialized training, rotational orientation deliberate training must be increased to enhance athletes' map representation ability, further improve cognitive ability, and promote the growth of key orienteering skills, such as map literacy and mental rotation ability.

## CONCLUSIONS

The study examined the effects of changes in mental rotation on the behavioral performance of map representation and prefrontal activation in orienteering athletes by using an fNIRS instrument and revealed that with the rotation of orientation, the behavioral performance of map representation decreased, and the functional activation of the dorsal lateral prefrontal and ventral lateral prefrontal regions appeared to be significantly more important. It was also found that under the rotational condition, more brain regions participated in the cognitive task, and the synergy of each brain region was enhanced, providing theoretical support for the construction of a model for scientific training and promoting the improvement of wayfinding and navigation ability in orienteering sports.

## ACKNOWLEDGEMENTS

The authors would like to thank the portable functional near-infrared spectroscopic imager (LIGHTNIRS system) manufactured by Shimadzu Corporation, Japan, for providing equipment support. The authors would also like to thank all the orienteering team athletes who participated in the experiments and the students who assisted in the experiments.

### Funding

This work was supported by the Social Science Fund Project of Shaanxi Province (2022P003) and High-level Achievement Cultivation Project of Shaanxi Normal University (2022BA002). The funders had no role in study design, data collection and analysis, decision to publish, or preparation of the manuscript.

### Grant Disclosures

The following grant information was disclosed by the authors:
Social Science Fund Project of Shaanxi Province: 2022P003.
High-level Achievement Cultivation Project of Shaanxi Normal University: 2022BA002.

### Competing Interests

The authors declare that they have no competing interests.

### Author Contributions

- Mingsheng Zhao conceived and designed the experiments, performed the experiments, analyzed the data, prepared figures and/or tables, authored or reviewed drafts of the article, and approved the final draft.
- Jingru Liu performed the experiments, analyzed the data, prepared figures and/or tables, and approved the final draft.
- Yang Liu conceived and designed the experiments, analyzed the data, authored or reviewed drafts of the article, and approved the final draft.
- Pengyang Kang conceived and designed the experiments, performed the experiments, analyzed the data, prepared figures and/or tables, and approved the final draft.

### Human Ethics

The following information was supplied relating to ethical approvals (*i.e.*, approving body and any reference numbers):

The Shaanxi Normal University granted Ethical approval to carry out the study within its facilities (Ethical Application Ref: SNNU2023301).

### Data Availability

The raw measurements are available in the Supplemental File.

## Supplemental Information

Supplemental information for this article can be found online at http://dx.doi.org/10.7717/peerj.16299#supplemental-information.

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
