# Peer review of "Effects of mental rotation on map representation in orienteers—behavioral and fNIRS evidence"

_PeerJ, doi:10.7717/peerj.16299_

## Round 0.1 · original submission · Major Revisions

As you can see, both Reviewers have major concerns with your paper that would require a major revision. Final acceptance of the manuscript is not guaranteed at this stage.

**Language Note:** The review process has identified that the English language must be improved. PeerJ can provide language editing services - please contact us at copyediting@peerj.com for pricing (be sure to provide your manuscript number and title). Alternatively, you should make your own arrangements to improve the language quality and provide details in your response letter. – PeerJ Staff

Reviewer 1 ·

Basic reporting

The manuscript titled ‘Effects of mental rotation on map representation in orienteers - Evidence from behavioral and fNIRS’ presents a study investigating the changes in the behavioral performance of orienteers during rotational orientation and the involvement of the ventral prefrontal cortex in map representation. While the study offers theoretical support for scientific training in orienteering sports programs, there are several concerns that need to be addressed before recommending acceptance.

The introduction lacks clarity in addressing the novelty of the study and establishing clear research objectives. It is important for the authors to provide a stronger rationale for their investigation and present a comprehensive background on the existing knowledge gaps. Furthermore, the hypothesis in the manuscript is not well-developed and lacks novelty. The authors should clearly articulate the unique aspects of their study and explain how it contributes to the current understanding of the subject matter.

Experimental design

Several methodological concerns were also identified. The use of a t-test instead of an ANOVA to detect potential differences between behavioral indicators under different rotational orientations requires further justification. Additionally, critical details such as the Shapiro-Wilk test value and degrees of freedom were omitted in the behavioral results. The type of analysis used in the fNIRS results was also not explicitly mentioned, leading to ambiguity.

Within the correlation analysis section, the explanation of how a person's correlation works is redundant and unnecessary. The authors should focus on the key findings and their implications instead of providing elementary statistical explanations.

Validity of the findings

In the discussion section, the authors fail to adequately explain the novel aspects of their study and the necessity of its findings. It is important to highlight the study's contributions to the field and discuss potential applications or implications of the results.

Additional comments

The figure numbering is inconsistent between the figures and the corresponding text. To improve organization, it is suggested to combine Figures 1 and 2, as well as Figures 6, 7, 8, and Figures 9 and 10. Furthermore, the figure captions do not provide sufficient information to understand the figures, requiring clearer and more comprehensive explanations. I also suggest improving the Figure's quality because the resolution is quite low.

Throughout the manuscript (title included), sentences are overly long and convoluted, leading to confusion. It is recommended to simplify and clarify the language to enhance readability. Furthermore, thorough proofreading is necessary to address the numerous grammatical errors and typos present in the manuscript.

Reviewer 2 ·

Basic reporting

- The paper needs to review the background on the correlates of map-use and the neural correlates of mental rotation, as well as map-use.

- There are a few errors in English in the abstract, introduction, and throughout.

Experimental design

- Although novelty is not necessarily a publication criterion, it is important to frame the findings with respect to the existing literature in order to understand its contribution. Was the study hypothesis-driven on a distinction between dorsal and ventral lateral PFC? Did it confirm previous results?

- The methods for preprocessing should be more detailed.l What was the low/high frequency filters on the signals?

- What is the basis for determining the PFC subregion localization and to what % is it accurate?

Validity of the findings

- Were the current results the same when analyzed on a channel by channel basis?

- How do we know the results are not a difficulty signal? Do they hold when comparing correct vs incorrect trials?

- It is not clear whether the study and the results are generally about map-use or whether the results are specific to orienteers. This needs to be discussed.

---

## Round 0.2 · Minor Revisions

Please address the further comments provided by the Reviewers.

Reviewer 1 ·

Basic reporting

Overall, I appreciate the authors' thorough responses to my previous feedback, and I believe that the revised manuscript has reached a state suitable for publication. However, there are still some minor issues that require attention:


1. At lines 57, 59 and 103, references are missing. The authors refer to specific studies not cited in the manuscript.

2. Line 113, please consider revising the sentence to: "To address the above-listed issues…."

3. On line 130, please consider revising the sentence to: "The informed consent of all participants was obtained and documented (see supplementary materials)."

4. Line 171: Here, authors should provide information on the duration of the familiarization phase.

5. Line 279: It should read "Fig. 6" instead of "Fig. 7."

Experimental design

no comment

Validity of the findings

no comment

Reviewer 2 ·

Basic reporting

I appreciated that the authors seem to have gotten their English edited by a professional service provider.

Experimental design

no comment

Validity of the findings

Given that this study does not track the change in neural activation with number of years or practice orienteering, the authors should be careful in their wording, avoiding asserting that there was a change brought on by orienteering (e.g., it could be that people who engage in orienteering were different to begin with).

---

## Round 0.3 · accepted · Accept

It seems to me that you have addressed satisfactorily all the reviewers' comments, and I am therefore happy to accept the paper for publication.